# NysReg-Gradient: Regularized Nyström-Gradient for Large-Scale Unconstrained Optimization and its Application

## Abstract

We develop a regularized Nyström method for solving unconstrained optimization problems with high-dimensional feature spaces. While the conventional second-order approximation methods such as quasi-Newton methods rely on the first-order derivatives, our method leverages the actual Hessian information. Additionally, Newton-sketch based methods employ a sketch matrix to approximate the Hessian, such that it requires the thick embedding matrix with a large sketch size. On the other hand, the randomized subspace Newton method projects Hessian onto a lower dimensional subspace that utilizes limited Hessian information. In contrast, we propose a balanced approach by introducing the regularized Nyström approximation. It leverages partial Hessian information as a thin column to approximate the Hessian. We integrate approximated Hessian with gradient descent and stochastic gradient descent. To further reduce computational complexity per iteration, we compute the inverse of the approximated Hessian-gradient product directly without computing the inverse of the approximated Hessian. We provide the convergence analysis and discuss certain theoretical aspects. We provide numerical experiments for strongly convex functions and deep learning. The numerical experiments for the strongly convex function demonstrate that it notably outperforms the randomized subspace Newton and the approximation of Newton-sketch which shows the considerable advancements in optimization with high-dimensional feature space. Moreover, we report the numerical results on the application of brain tumor detection, which shows that the proposed method is competitive with the existing quasi-Newton methods that showcase its transformative impact on tangible applications in critical domains.

## 1 Introduction

The optimization of various functions is a crucial and highly relevant topic in machine learning, particularly due to the exponential growth in data volume. As a result, finding solutions to large-scale optimization problems has become a pressing concern. In this paper, we propose a method to address this challenge by approximating the Hessian matrix of the objective function using the Nyström approximation. Our approach aims to solve a large-scale unconstrained optimization problem of the form:

$$\min_{\boldsymbol{w} \in \mathbb{R}^d} f(\boldsymbol{w}) \tag{1}$$

where $f$ is twice continuously differentiable and $f$ is convex.

The traditional second-order optimizers to solve (1), such as Newton's method provide quadratic convergence. However, these methods face limitations when dealing with high-dimensional optimization problems due to their high per-iteration cost and memory requirements. To address this challenge, we provide a low-rank Hessian approximation method that iteratively uses the Nyström method or more generally, a column subset selection method to approximate the Hessian. By employing this approach, we aim to provide a computationally efficient alternative that overcomes the limitations of traditional second-order optimizers for high-dimensional problems.

## 1.1 Background and contributions

To optimize (1), first-order optimization methods such as stochastic gradient descent (SGD) (Robbins & Monro, 1951), AdaGrad, stochastic variance-reduced gradient (SVRG) (Johnson & Zhang, 2013), Adam (Kingma & Ba, 2015), and the stochastic recursive gradient algorithm (SARAH), possibly augmented with momentum, are preferred for large-scale optimization problems owing to their more affordable computational costs, which are linear in dimensions per epoch $O(nd)$. However, the convergence of the first-order methods is notably slow, and they are sensitive to hyperparameter choices and ineffective for ill-conditioned problems.

In contrast, Newton's method does not depend on the parameters of specific problems and requires only minimal hyperparameter tuning for self-concordant functions, such as $\ell_2$-regularized logistic regression. However, Newton's method involves a computational complexity of $\Omega(nd^2 + d^{2.37})$ (Agarwal et al., 2017) per iteration and thus is not suitable for large-scale settings. To reduce this computational complexity, the subsampled Newton's method and random projection (or sketching) are commonly used to reduce the dimensionality of the problem and solve it in a lower-dimensional subspace. The subsampled (a.k.a mini-batch) Newton method performs well for large-scale but relatively low-dimensional problems by computing the Hessian matrix on a relatively small sample. However, it is time-consuming for high-dimensional problems. Randomized algorithms (Lacotte et al., 2021; Pilanci & Wainwright, 2017) estimate the Hessian in Newton's method using a random embedding matrix $\boldsymbol{S} \in \mathbb{R}^{m \times n}$, $\boldsymbol{H_S}(\boldsymbol{w}) := (\nabla^2 f(\boldsymbol{w})^{\frac{1}{2}})^\top \boldsymbol{S}^\top \boldsymbol{S} (\nabla^2 f(\boldsymbol{w})^{\frac{1}{2}})$. Specifically, their approximation used the square root of the generalized Gauss-Newton (GGN) matrix as a low-rank approximation instead of deriving it from actual curvature information, whereas $\boldsymbol{S}$ is a random projection matrix of size $(m \times n)$. Moreover, the Newton sketch Pilanci & Wainwright (2017) requires a substantially large sketch size which can be as big as the dimension $d$, which is not ideal and over-matches the objective of a low-rank Hessian approximation.

Recently, Derezinski et al. (2021) proposed the Newton-LESS method which is based on the leverage score specified embeddings. It sparsified the Gaussian sketching and reduced the computational cost with similar convergence properties as the dense Gaussian sketching.

Gower et al. (2019) proposed the randomized subspace Newton (RSN) method. RSN is the randomized subspace Newton that computes the sketch of Hessian by sampling the embedding matrix $\boldsymbol{S}$ and approximating the Hessian as $\boldsymbol{S}(\boldsymbol{S}^\top \boldsymbol{H} \boldsymbol{S})^\dagger \boldsymbol{S}^\top$.

Talwalkar (2010) proposed the Nyström logistic regression algorithm, where the Nyström method is used to approximate the Hessian of the regularized logistic regression. Thus, it can be regarded as a variant of Nyström-SGD. However, Talwalkar (2010) only considered the regularized logistic regression, in which the Hessian can be explicitly obtained, with deterministic optimization. In contrast, we propose the regularized Nyström method for the deterministic and stochastic optimization, such that the value of the regularizer depends on the norm of gradient or stochastic gradient, respectively. We also show its theoretical aspects in terms of rank and no. of randomly picked columns.

The limited-memory Broyden-Fletcher-Goldfarb-Shanno (LBFGS) algorithm (Liu & Nocedal, 1989) is a widely used quasi-Newton method. More specifically, it estimates the Hessian inverse using the past difference of gradients and updates. The online BFGS (oBFGS) (Schraudolph et al., 2007) method is a stochastic version of regularized BFGS and L-BFGS with gradient descent. Kolte et al. (2015) proposed two variants of a stochastic quasi-Newton method incorporating a variance-reduced gradient. The first variant used a sub-sampled Hessian with singular value thresholding. The second variant used the LBFGS method to approximate the Hessian inverse. The stochastic quasi-Newton method (SQN) (Byrd et al., 2016) used the Hessian vector product computed on a subset of each mini-batch instead of approximating the Hessian inverse from the difference between the current and previous gradients, as in LBFGS. SVRG-SQN (Moritz et al., 2016) also incorporated variance-reduced gradients.

**Contributions:** The contributions of this study are summarized as follows.

- We propose the (deterministic and stochastic) regularized Nyström approximated Hessian method to solve the unconstrained optimization problem.

- We propose to use the regularizer obtained by gradient information to regularize the Nyström approximation.

- We provide detailed proof of the convergence. Moreover, we present various theoretical aspects of the proposed method.

- We empirically show numerical experiments of the proposed methods and compare them with those of existing methods on the benchmark datasets.

- In addition, we consider a classification problem of tumor detection as an application for Brain MRI and show the performance of the proposed method by comparing it with similar existing methods.

## 2 Nyström approximation and its properties

When dealing with large datasets, the computational complexity of second-order optimization methods poses a significant challenge. As a result, there is a need to explore computationally feasible Hessian approximation techniques that offer theoretical guarantees. Over the past few decades, researchers have investigated various matrix approximation methods. In recent years, a common approach involves obtaining a low-rank approximation of a matrix by utilizing specific parts of the original matrix through various techniques. One popular method in this context is the Nyström approximation (Drineas & Mahoney, 2005), initially introduced for kernel approximation. The Nyström approximation is a low-rank approximation of a positive semidefinite matrix that leverages partial information from the original matrix to construct an approximate matrix of lower rank. The Nyström method can be categorized as a variant of the column subset selection problem. Talwalkar (Talwalkar & Rostamizadeh, 2014) proposed minimizing the error using low-coherence bounds of the Nyström method. Michel Derezinski (Derezinski et al., 2020) proposed improvements in the approximation guarantees of column subset selection and the Nyström method using spectral properties.

**Definition 1** (Nyström approximation). Let $\boldsymbol{H} \in \mathbb{R}^{d \times d}$ be a symmetric positive semi-definite matrix. Then, choose $m$ columns of $\boldsymbol{H}$ randomly to form a $d \times m$ matrix $\boldsymbol{C}$. Let $m \times m$ be a matrix $\boldsymbol{M}$ such that it is formed by the intersection of those $m$ columns and corresponding $m$ rows of $\boldsymbol{H}$. $\boldsymbol{M}_k$ is the best $k$-rank approximation of $\boldsymbol{M}$. A $k$-rank Nyström approximation $\boldsymbol{N}_k$ of $\boldsymbol{H}$ can be defined as

$$\boldsymbol{N}_k = \boldsymbol{C}\boldsymbol{M}_k^\dagger \boldsymbol{C}^\top. \tag{2}$$

where $\boldsymbol{M}_k^\dagger$ is a pseudo-inverse of $\boldsymbol{M}_k$. Letting $\boldsymbol{H} = \nabla^2 f(\boldsymbol{w})$ to be a Hessian matrix of the objective function (1), following theorem shows the distance between the Hessian $\boldsymbol{H}$ and the Nyström approximation $\boldsymbol{N}$ of $\boldsymbol{H}$.

**Theorem 1.** *(Drineas & Mahoney, 2005, Algorithm 2) Let $\boldsymbol{H}$ be a $d \times d$ matrix and let $\boldsymbol{N}_k = \boldsymbol{C}\boldsymbol{M}_k^\dagger \boldsymbol{C}^\top$ be a $k$-rank ($k \leq m$) is a Nyström approximation by sampling $m$ columns of $\boldsymbol{H}$ with probabilities $\{p_i\}_{i=1}^d$ such that*

$$p_i = \frac{\boldsymbol{H}_{ii}^2}{\sum_{i=1}^d \boldsymbol{H}_{ii}^2}. \tag{3}$$

*Let $k = rank(\boldsymbol{M})$ and let $\boldsymbol{H}_k$ be the best $k$-rank approximation of the $\boldsymbol{H}$. In addition, let $\varepsilon > 0$ and $\vartheta = 1 + \sqrt{8 \log(1/\varrho)}$. If (a) $m \geq 64k\vartheta^2/\varepsilon^4$, (b) $m \geq 4\vartheta^2/\varepsilon^4$, then with probability at least $1 - \varrho$*

$$\|\boldsymbol{H} - \boldsymbol{N}_k\|_\nu \leq \|\boldsymbol{H} - \boldsymbol{H}_k\|_\nu + \varepsilon \sum_{i=1}^d \boldsymbol{H}_{ii}^2, \tag{4}$$

*for (a) $\nu = F$ (Frobenius) and (b) $\nu = 2$ (spectral), where $\varepsilon > 0$.*

We denote above upper bound as $U_{Nys} = \|\boldsymbol{H} - \boldsymbol{H}_k\|_\nu + \varepsilon \sum_{i=1}^d \boldsymbol{H}_{ii}^2$ for the rest of paper.

An alternative way to define a $k$-rank Nyström approximation is via zero-one sampling matrix. Let $\boldsymbol{H} = \nabla^2 f(\boldsymbol{w})$ be a Hessian of $f(\boldsymbol{w})$ that has form of $\boldsymbol{H} = \boldsymbol{X}^\top \boldsymbol{X}$, where $\boldsymbol{X}$ is an $n \times d$ matrix. If is always possible

to assume that $\boldsymbol{H} = \boldsymbol{X}^\top \boldsymbol{X}$ because $\boldsymbol{H}$ is a symmetric positive semi-definite (SPSD). The zero-one matrix $\boldsymbol{W} \in \mathbb{R}^{d \times m}$ can be constructed as follows.

$$\boldsymbol{W}(i,j) = \begin{cases} 1 & \text{if the } i\text{-th column is chosen in} \\ & \text{the } j\text{-th random trail,} \\ 0 & \text{otherwise.} \end{cases} \tag{5}$$

We can write the Nystöm approximation using zero-one matrix as follows:

$$\boldsymbol{C}(\boldsymbol{M}_k)^\dagger \boldsymbol{C}^\top = (\boldsymbol{H}\boldsymbol{W})(\boldsymbol{W}^\top \boldsymbol{H}\boldsymbol{W})_k^\dagger (\boldsymbol{H}\boldsymbol{W})^\top. \tag{6}$$

Drineas & Mahoney (2005) shows that the uniform sampling case of scaled Nyström brings the same expression as the (6). It can be defined as follows:

$$\boldsymbol{C}(\boldsymbol{M}_k)^\dagger \boldsymbol{C}^\top = (\boldsymbol{H}\boldsymbol{W}\boldsymbol{D})((\boldsymbol{W}\boldsymbol{D})^\top \boldsymbol{H}\boldsymbol{W}\boldsymbol{D})_k^\dagger (\boldsymbol{H}\boldsymbol{W}\boldsymbol{D})^\top$$

where $\boldsymbol{D} \in \mathbb{R}^{m \times m}$ is a scaling matrix that have diagonal entries $1/\sqrt{mp_{i_l}}$, $p_{i_l}$ is a probability $\mathbf{P}(i_l = i) = p_i$ given in (3) of the Theorem 1 and $i_l$ is a column chosen in $l$th independent trail. Moreover, $\boldsymbol{C} := \boldsymbol{H}\boldsymbol{W}$, which is the sampled column matrix of the true Hessian, and $\boldsymbol{M} := \boldsymbol{W}^\top \boldsymbol{H}\boldsymbol{W}$, which is the intersection matrix in (2). However, if we let $m = k$ and then in the case of uniform sampling, the probability $p_i = 1/d$, and scaling matrix have diagonal entries $D_{ii} = \sqrt{\frac{d}{m}}$ which obtain the approximation (2) that is exactly same as the (6).

*Remark* 1. Consider an instance of a function $f(\boldsymbol{w}) = \ell(\boldsymbol{A}\boldsymbol{w})$, where $\boldsymbol{A} \in \mathbb{R}^{n \times d}$ and $\ell : \mathbb{R}^n \to \mathbb{R}$ has separable form such that $\ell(\boldsymbol{A}\boldsymbol{w}) = \sum_{i=1}^n \ell_i(\langle \boldsymbol{a}_i, \boldsymbol{w} \rangle)$ the square-root of Hessian can be computed as $\boldsymbol{X}^\top = \nabla^{1/2} f(\boldsymbol{w}) = \text{diag}\{\ell_i''\}_{i=1}^n A$.

Let $\boldsymbol{S} = \boldsymbol{W}\boldsymbol{D}$ and one can compute Nyström approximation using $\boldsymbol{S}$. However, generalized Nyström method analyzed in Frangella et al. (2021); Gittens (2011); Tropp et al. (2017) consider the theory with the the Gaussian and various interesting random matrices $\boldsymbol{S}$. Therefore, we also consider a Gaussian random matrix.

**Lemma 1.** *Fuji et al. (2022) Let $\boldsymbol{S}$ be a $d \times m$ random matrix such that $s_{ij}$ are independently sampled from the normal distribution $\mathrm{N}(0, 1/\mathrm{m})$, then there exists $\mathcal{C} > 0$ such that*

$$\|\boldsymbol{S}^\top \boldsymbol{S}\| \le \mathcal{C}\frac{d}{m}.$$

*with probability at least $1 - 2\exp(-m)$, where $\mathcal{C}$ is an absolute constant.*

One can prove above lemma from the (Vershynin, 2018, Theorem 4.6.1). For the rest of theoretical analysis, we consider the matrix $\boldsymbol{S}$ to be a generalized random matrix given in Lemma 1.

## 3 Algorithmic framework

In this section, we first define a formulation of the Nyström approximation for the objective function (1) and propose the regularized Nyström algorithm for the unconstrained optimization problem.

Let $\boldsymbol{H} = \nabla^2 f(\boldsymbol{w})$ be a Hessian of the objective function, and we pick $\boldsymbol{\Omega} \subseteq \{1, 2, \ldots, d\}$ indices uniformly at random such that $m = |\boldsymbol{\Omega}|$ and compute the Nyström approximation as

$$\boldsymbol{N}_k = \boldsymbol{C}\boldsymbol{M}_k^\dagger \boldsymbol{C}^\top = \boldsymbol{Z}\boldsymbol{Z}^\top, \tag{7}$$

where $\boldsymbol{Z} = \boldsymbol{C}\boldsymbol{U}_k \boldsymbol{\Sigma}_k^{-1/2} \in \mathbb{R}^{d \times k}$, and $\boldsymbol{C} \in \mathbb{R}^{d \times m}$ is a matrix consisting of $m$ columns ($m \ll d$) of $\boldsymbol{H}$, $\boldsymbol{M}$ is $m \times m$ intersection matrix, and the rank of $\boldsymbol{M}$ is $k \le m$. We obtain the best $k$ rank approximation using the singular value decomposition (SVD) of $\boldsymbol{M}_k$ as $\boldsymbol{M}_k = \boldsymbol{U}_k \boldsymbol{\Sigma}_k \boldsymbol{U}_k^\top$, where $\boldsymbol{U}_k \in \mathbb{R}^{m \times k}$ are singular vectors and $\boldsymbol{\Sigma}_k \in \mathbb{R}^{k \times k}$ consisting $k$ singular values. The pseudo-inverse can be computed as $\boldsymbol{M}_k^\dagger = \boldsymbol{U}_k \boldsymbol{\Sigma}_k^{-1} \boldsymbol{U}_k^\top$. Note that the number of columns $m$ is a hyperparameter.

### 3.1 Relation between $\ell_2$ regularization and fixed rank Nyström approximation

Consider $\ell_2$ regularized objective function

$$\min_{\boldsymbol{w}\in\mathbb{R}^d}\left\{f(\boldsymbol{w}):=\sum_{i=1}^n f_i(\boldsymbol{w})+\frac{\lambda}{2}\|\boldsymbol{w}\|^2\right\}, \tag{8}$$

where each $f_i$ are convex, twice continuously differentiable, and $\lambda \geq 0$, and hence $f$ is strongly convex function. Then the Hessian of $\ell_2$-regularized function can be given as $\boldsymbol{H} = \sum_{i=1}^n \nabla^2 f_i(\boldsymbol{w}) + \lambda \boldsymbol{I}$ and $\lambda_{\min}(f(\boldsymbol{w})) \geq \lambda$. The formulation of column matrix $\boldsymbol{C} = \boldsymbol{S}^\top \left(\sum_{i=1}^n \nabla^2 f_i(\boldsymbol{w})\right) + \lambda \boldsymbol{S}^\top \boldsymbol{I}$ and matrix $\boldsymbol{M} = \boldsymbol{S}^\top \left(\sum_{i=1}^n \nabla^2 f_i(\boldsymbol{w})\right) \boldsymbol{S} + \lambda \boldsymbol{S}^\top \boldsymbol{S} \in \mathbb{R}^{m\times m}$. Since $\lambda$ is used in the approximation, matrix $M$ becomes positive definite and hence it becomes the fixed ranked Nyström approximation, which also helps in the convergence to get minimum eigenvalue of $M^{-1}$. Hence, we can write it as

$$\boldsymbol{N} = \boldsymbol{C}\boldsymbol{M}^{-1}\boldsymbol{C}^\top$$

for fixed rank Nyström approximation.

## 4 NysReg-gradient: Regularized Nyström-gradient method

Second-order optimization methods often utilize the regularized approximated Hessian. Regularized parameters can be obtained through approaches such as the trust-region method or by adaptively determining the regularization parameter based on the gradient information. These approaches have been explored in previous works such as Li et al. (2004); Ueda & Yamashita (2010); Tankaria et al. (2022), which propose iterative formulations similar to:

$$\boldsymbol{w}_{t+1} = \boldsymbol{w}_t - \eta_t(\boldsymbol{A}_t + \rho_t \mathbf{I})^{-1}\nabla f(\boldsymbol{w}_t), \tag{9}$$

where, $\boldsymbol{A}_t$ represents a Hessian approximation, and $\rho_t > 0$ is a regularized parameter.

Now, consider $\boldsymbol{A}_t$ to be Nyström approximation $\boldsymbol{N}_t$ in equation (9). To ensure the non-singularity and obtain a descent direction, we compute a regularized Nyström approximation. Then we can write an iterate of the regularized Nyström approximation as

$$\boldsymbol{w}_{t+1} = \boldsymbol{w}_t - \eta_t(\boldsymbol{N}_t + \rho_t \mathbf{I})^{-1}\nabla f(\boldsymbol{w}_t). \tag{10}$$

Since we are approximating the Hessian using Nyström method augmented with a regularizer in a similar quasi-Newton framework that uses the multiple of gradient in the search direction, we call our novel method "NysReg-gradient: Regularized Nyström gradient method (NGD)". The regularized parameter $\rho_t > 0$ is determined based on the gradient information. Specifically, we set $\rho_t = c_1\|\nabla f(\boldsymbol{w}_t)\|^\gamma$ as similar to Ueda & Yamashita (2010), where $c_1 > 0$. We consider $\rho_t$ to be either $c_1\sqrt{\|\nabla f(\boldsymbol{w}_t)\|}$ for $\gamma = 1/2$, $c_1\|\nabla f(\boldsymbol{w}_t)\|$ for $\gamma = 1$, or $c_1\|\nabla f(\boldsymbol{w}_t)\|^2$ for $\gamma = 2$ as shown in Table 1. We denote $\nabla f(\boldsymbol{w}_t) = \boldsymbol{g}_t$ for the rest of paper.

Table 1: Relation between proposed methods and value of $\gamma$

| Proposed methods | Value of $\gamma$ | Regularizer $\rho_t$ | Regularized Nyström |
|---|---|---|---|
| NGD | $\gamma = 1/2$ | $\rho_t = c_1\|\boldsymbol{g}_t\|^{1/2}$ | $\boldsymbol{N}_t + c_1\|\boldsymbol{g}_t\|^{1/2}$ |
| NGD1 | $\gamma = 1$ | $\rho_t = c_1\|\boldsymbol{g}_t\|^1$ | $\boldsymbol{N}_t + c_1\|\boldsymbol{g}_t\|^1$ |
| NGD2 | $\gamma = 2$ | $\rho_t = c_1\|\boldsymbol{g}_t\|^2$ | $\boldsymbol{N}_t + c_1\|\boldsymbol{g}_t\|^2$ |

To efficiently compute the inverse of $(\boldsymbol{N}_t + \rho_t)$ given in (10), we use the Sherman–Morrison–Woodbury identity as

$$\boldsymbol{p}_t = (\boldsymbol{N}_t + \rho_t \boldsymbol{I})^{-1}\boldsymbol{g}_t = \frac{1}{\rho_t}\boldsymbol{g}_t - \boldsymbol{Q}_t\boldsymbol{Z}_t^\top \boldsymbol{g}_t, \tag{11}$$

where $\boldsymbol{p}_t$ is search direction at $t$th iteration, $\boldsymbol{N}_t$ is Nyström approximation computed at $\boldsymbol{w}_t$, $\boldsymbol{g}_t$ is a gradient computed at $\boldsymbol{w}_t$ and $\boldsymbol{Q}_t = \frac{1}{\rho_t^2}\boldsymbol{Z}_t(\boldsymbol{I}_k + \frac{1}{\rho_t}\boldsymbol{Z}_t^\top\boldsymbol{Z}_t)^{-1}$. Here, $(\boldsymbol{I}_k + \frac{1}{\rho_t}\boldsymbol{Z}_t\boldsymbol{Z}_t^\top) \in \mathbb{R}^{k\times k}$, and its inverse can be computed much more quickly than the inverse of $(\boldsymbol{N}_t + \rho_t\boldsymbol{I})$ directly. We use the backtracking line search with Armijo's line search rule that finds a step size $\eta_t = \alpha^{(\ell)} = \tau\alpha^{(\ell-1)}$, starting from $\ell = 0$, the initial step size $\eta_0 = \alpha^{(0)}$, and finds the least positive integer $\ell \geq 0$ and increased $\ell$ by $\ell + 1$ until the

$$f(\boldsymbol{w}_t + \alpha^{(\ell)}\boldsymbol{p}_t) \leq f(\boldsymbol{w}_t) + \alpha^{(\ell)}\beta\boldsymbol{g}_t^\top\boldsymbol{p}_t, \tag{12}$$

holds, where $\alpha, \beta \in (0,1)$. Next, we introduce the main algorithm.

---

**Algorithm 1** NysReg-gradient: Regularized Nyström-Gradient Algorithm

---

1: **Initialize** Initial parameters $\boldsymbol{w}_0$, desired rank $|\boldsymbol{\Omega}| = m$, $\alpha, \beta \in (0,1)$, and maximum iterations $t_{\max}$
2: $t \leftarrow 0$
3: **repeat**
4:     $\boldsymbol{g}_t = \nabla f(\boldsymbol{w}_t)$
5:     randomly pick indices set $\boldsymbol{\Omega} \subseteq \{1,2,\ldots,d\}$ such that $m = |\boldsymbol{\Omega}|$
6:     compute $\boldsymbol{C}_t$ ($\boldsymbol{\Omega}$ columns of the Hessian)
7:     compute $\boldsymbol{Z}_t$ using (7) and compute $\rho_t$
8:     $\boldsymbol{Q}_t = \frac{1}{\rho_t^2}\boldsymbol{Z}_t(\boldsymbol{I}_k + \frac{1}{\rho_t}\boldsymbol{Z}_t^T\boldsymbol{Z}_t)^{-1}$
9:     Compute $(\boldsymbol{N}_t + \rho_t\boldsymbol{I})^{-1}\boldsymbol{g}_t$ using (11)
10:    Use backtracking line search with Armijo's rule to find $\eta_t$ using (12)
11:    $\boldsymbol{w}_{t+1} = \boldsymbol{w}_t - \eta_t\boldsymbol{p}_t$
12:    $t = t + 1$
13: **until** $t = t_{\max}$ or some termination criteria is satisfied
14: **return** $\boldsymbol{w}_t$

---

The efficiency of the method depends on both rank of Hessian and the choice of the sketching matrix $S$. For example if the sketch size goes to one then method reduces to scaled gradient descent. Next, we see discuss the computational complexity of the proposed algorithm.

## 4.1 Computational complexity

Here, we analyze the per-iteration computational complexity of the proposed method. The cost of matrix-vector multiplication $(\boldsymbol{N}_t + \rho_t\boldsymbol{I})^{-1}\boldsymbol{g}_t$, *i.e.*, (11) is $O(dk)$ at each iteration. The cost of computing $\boldsymbol{Q}_t$ is $O(dk^2)$ at each epoch. The cost of computing $\boldsymbol{Z}_t$ is $O(dmk)$. The computational cost of constructing the matrix $\boldsymbol{C}$ is $O(dm)$. Thus, over the course of all iterations, the construction of the matrix $\boldsymbol{C}$ is associated with the highest computational cost; therefore, the overall time and space complexity are $O(dm)$.

## 4.2 Regularized Nyström as Newton sketch

In this section, we introduce an alternate definition of the Nyström approximation. Nyström approximation can be obtained by sampling the embedding (random sketch) matrix. We further show that resultant formulation of an alternate definition of the Nyström approximation and it can be interpreted as a Newton sketch-based method (Pilanci & Wainwright, 2017; Lacotte et al., 2021). Consider the Nyström approximation and let $\boldsymbol{H} = \boldsymbol{X}_{d\times n}^\top\boldsymbol{X}_{n\times d}$ and zero-one $d \times m$ matrix $\boldsymbol{W}$ in (5) with $\boldsymbol{C}_X = \boldsymbol{XW}$. Let SVD of $\boldsymbol{XW}$ is $\widehat{\boldsymbol{U}}\widehat{\boldsymbol{\Sigma}}\widehat{\boldsymbol{V}}^\top$, and $\boldsymbol{M} = (\boldsymbol{C}_X^\top\boldsymbol{C}_X) = \widehat{\boldsymbol{V}}\widehat{\boldsymbol{\Sigma}}^2\widehat{\boldsymbol{V}}^\top$. Then, similar to (Drineas & Mahoney, 2005, Lemma 4) we obtain,

$$\begin{aligned}
\boldsymbol{C}(\boldsymbol{M}_k)^\dagger\boldsymbol{C}^\top &= (\boldsymbol{HW})(\boldsymbol{W}^\top\boldsymbol{HW})_k^\dagger(\boldsymbol{HW})^\top \\
&= (\boldsymbol{X}^\top\boldsymbol{C}_X)(\boldsymbol{C}_X^\top\boldsymbol{C}_X)_k^\dagger(\boldsymbol{X}^\top\boldsymbol{C}_X)^\top \\
&= \boldsymbol{X}^\top(\widehat{\boldsymbol{U}}\widehat{\boldsymbol{\Sigma}}_k\widehat{\boldsymbol{V}}^\top)(\widehat{\boldsymbol{V}}\widehat{\boldsymbol{\Sigma}}_k^{-2}\widehat{\boldsymbol{V}}^\top)(\widehat{\boldsymbol{V}}\widehat{\boldsymbol{\Sigma}}_k\widehat{\boldsymbol{U}}^\top)\boldsymbol{X}. \\
&= \boldsymbol{X}^\top\widehat{\boldsymbol{U}}_k\widehat{\boldsymbol{U}}_k^\top\boldsymbol{X}
\end{aligned} \tag{13}$$

where $\widehat{U}_k$ is $k-$rank matrix. The right-hand side of (13) is similar to the Newton sketch Pilanci & Wainwright (2017) with two differences, **1)** embedding matrix $P$ depends on the size of $n$ and not $d$, whereas the zero-one matrix $W \in \mathbb{R}^{d \times m}$ depends on the $d$ and **2)** the natural orthogonal matrix $\widehat{U}_k$ in proposed method is replaced by a randomized embedding matrix $P^\top \in \mathbb{R}^{n \times m}$, which is expected to be orthogonal in principle. *i.e.,* $\mathbb{E}[P^\top P] = I$, whereas the proposed method produces the natural orthogonal matrix; *i.e.,* $\widehat{U}\widehat{U}^\top = I$. Consequently, Newton-sketch needs a large and thick column matrix $P$ (assuming most data having $n > d$) to approximate the Hessian.

If we let $X = \nabla^2 f(w)^{1/2}$ then, our approximation is of the form of

$$\begin{aligned} H_W &= X^\top \widehat{U}\widehat{U}^\top X + \lambda I \\ &= (\nabla^2 f(w)^{1/2})^\top \widehat{U}\widehat{U}^\top (\nabla^2 f(w)^{1/2}) + \rho I. \end{aligned} \tag{14}$$

More generally, the approximation given above can be written in the form of an embedding matrix as follows. Let $Y = \rho I_d$, and let $Y^{1/2} = \sqrt{\rho} \cdot I_d$ be a $d \times d$ matrix. Then, by defining the embedding matrix $\bar{S} = \begin{bmatrix} \widehat{U}^\top_{m \times n} & 0_{m \times d} \\ 0_{d \times n} & I_d \end{bmatrix}$ and partial Hessian $\bar{H} = \begin{bmatrix} \nabla^2 f(w)^{1/2} \\ Y^{1/2} \end{bmatrix}$, we get

$$H_S = \bar{H}^\top \bar{S}^\top \bar{S}\bar{H}$$

which is identical to the (14) and hence $H_S^{-1}$ is non-singular, where $H_S$ is the Nyström approximation for $H_S$. Note that $X = \nabla^2 f(w)^{1/2}$ can be computed as shown in the remark 1.

## 5 Convergence analysis

In this section, we provide the analysis that is based on selecting the number of columns $m$, in the Nyström approximation. We investigate distance between the Newton's direction and the NGD's search direction that is based on the rank of matrix $M$. We further prove the linear convergence of the proposed algorithm. Moreover, in the last subsection, we discuss the closeness of the inverse of regularized Nyström with the inverse of Hessian. This analysis offers insights into the overall convergence behavior of the algorithm. It is important to note that our convergence analysis is based on the objective function defined in equation (8).

For local convergence, see section A given in the Appendix.

Next, we provide the convergence analysis. First, we need following assumptions.

**Assumption 1.** *i)* The objective function (8) is twice continuously differentiable and $f$ is $L_g$-smooth, *i.e.,*

$$\|\nabla^2 f(w_t)\| \leq L_g, \quad \forall\ w_t \in \mathbb{R}^d. \tag{15}$$

*ii)* The objective function (8) is strongly convex.

**Assumption 2.** $S_t$ is a random matrix whose entries are independently sampled Normal distribution with mean 0 and variance $1/m$, satisfies

$$\|S^\top S\| \leq \mathcal{C}\frac{d}{m},$$

for some $\mathcal{C} > 0$.

**Assumption 3.** For dimension $d$, we have a constraint on the value of $m$ such that $m = o(d)$.

Note that Assumption 3 is important as in the case where $m = d$, the Nyström approximation results into the Hessian, *i.e.,* $HH^\dagger H = H$ and it turns out to be the Newton's method.

In the next Lemma, we obtain lower bound of *minimum* eigenvalue and upper bound of *maximum* eigenvalue of $(N_t + \rho_t I)^{-1}$.

**Lemma 2.** *Suppose that Assumption 1, and 2 hold. Let $w_t$ iterate obtained by Algorithm 1, and for some $m$, the maximum and minimum eigenvalues of $(N_t + \rho_t I)^{-1}$ are given as*

$$\lambda_{\min}[(N_t + \rho_t I)^{-1}] \geq \frac{1}{\frac{\mathcal{C}L_g^2 d}{m\lambda} + c_1\|g_t\|^\gamma} \quad and \quad \lambda_{\max}[(N_t + \rho_t I)^{-1}] = \frac{1}{c_1\|g_t\|^\gamma}. \tag{16}$$

*Proof.* First we obtain the bound on *minimum* eigenvalue of $(\boldsymbol{N}_t + \rho\boldsymbol{I})^{-1}$.

$$
\begin{aligned}
\lambda_{\min}[(\boldsymbol{N}_t + \rho_t\boldsymbol{I})^{-1}] &= \frac{1}{\lambda_{\max}(\boldsymbol{N}_t + \rho_t\boldsymbol{I})} \\
&\geq \frac{1}{\lambda_{\max}(\boldsymbol{H}_t\boldsymbol{S}_t(\boldsymbol{S}^\top\boldsymbol{H}_t\boldsymbol{S}_t^\dagger)\boldsymbol{S}^\top\boldsymbol{H}_t) + \rho_t\boldsymbol{I}} \\
&\geq \frac{1}{\|\boldsymbol{H}_t\|^2\|\boldsymbol{S}_t^\top\boldsymbol{S}_t\|\|(\boldsymbol{S}^\top\boldsymbol{H}_t\boldsymbol{S}_t)^{-1}\| + \rho_t} \\
&\geq \frac{1}{L_g^2\left(\frac{\mathcal{C}d}{m}\right)\left(\frac{1}{\lambda}\right) + \rho_t} \\
&= \frac{1}{\frac{\mathcal{C}L_g^2 d}{m\lambda} + c_1\|\boldsymbol{g}_t\|^\gamma}
\end{aligned}
\tag{17}
$$

where the third inequality follows from the $\|\boldsymbol{H}\| \leq L_g$, Lemma 1, and since $f$ is strongly convex and $m \ll d$, $(\boldsymbol{S}^\top\boldsymbol{H}\boldsymbol{S}) \succeq \lambda\boldsymbol{I}$. Now we find obtain the bound on *maximum* eigenvalue of $(\boldsymbol{N}_t + \rho\boldsymbol{I})^{-1}$.

$$
\begin{aligned}
\lambda_{\max}[(\boldsymbol{N}_t + \rho_t\boldsymbol{I})^{-1}] &= \frac{1}{\lambda_{\min}(\boldsymbol{N}_t + \rho_t\boldsymbol{I})} \\
&= \frac{1}{\rho_t} \\
&= \frac{1}{c_1\|\boldsymbol{g}_t\|^\gamma}
\end{aligned}
$$

Since $\boldsymbol{N}_t$ is positive semi-definite $\rho_t$ is *minimum* eigenvalue of $(\boldsymbol{N}_t + \rho_t\boldsymbol{I})$. This completes the proof. □

## 5.1 Exactness of Nyström approximation

Here, we present a result to obtain the the distance between Hessian and Nyström approximation based on the size of the number of columns $m$ or rank of $M$. Kumar et al. (2009) showed a stronger result in the following theorem that if the rank of $\boldsymbol{M}$ is the same as the rank of $\boldsymbol{H}$, then Nyström approximation is exact.

**Theorem 2.** *(Kumar et al., 2009, Theorem 3) Suppose $\boldsymbol{H} \in \mathbb{R}^{d \times d}$ is positive semi-definite matrix and $rank(\boldsymbol{H}) = r \leq d$. Consider the Nyström approximation $\boldsymbol{N} = \boldsymbol{C}\boldsymbol{M}^\dagger\boldsymbol{C}^\top$ and $rank(\boldsymbol{M}) = r \leq m \leq d$, where $m$ is the number of columns picked randomly. Then the Nyström approximation is exact. i.e.,*

$$
\|\boldsymbol{H} - \boldsymbol{N}\|_F = 0,
$$

*where $\|.\|_F$ is the Frobenious norm.*

Note that $\|\boldsymbol{A}\|_2 \leq \|\boldsymbol{A}\|_F$ for any matrix $\boldsymbol{A}$. From the above theorem, it is easy to see that Nyström approximation produces the exactly same singular values when rank$(\boldsymbol{M}) = r$. Hence we can expect to achieve the same convergence as the Newton's method or superlinear convergence at least when the number of columns chosen $m \geq r$ and rank$(\boldsymbol{H}) = $ rank$(\boldsymbol{M}) = r$. Moreover, it tells that when rank$(\boldsymbol{M}) < r$, then we can not achieve the quadratic convergence since the distance between Hessian and Nyström is bounded from above and not exactly zero.

*Remark* 2. To have the least possible value of $m$ (*i.e.*, $m = r$) that satisfies the above theorem, we need to choose exactly those $r$ independent columns of $\boldsymbol{H}$ which is difficult due to the randomness involved in choosing $m$. In short, when rank$(\boldsymbol{H}) = r$, it becomes a feature selection problem to choose the $r$ independent columns that will form a Nyström approximation. The usual convergence gives a probabilistic convergence due to the randomness involved in $m$ and the convergence rate depends on the size of the number of randomly chosen columns $m = |\Omega|$.

## 5.2 Bound on the difference between NysReg-gradient's and Newton's direction

**Assumption 4.** For all $\boldsymbol{x}, \boldsymbol{y}$, the gradient is Lipschitz, *i.e.*,

$$\|\nabla f(\boldsymbol{x}) - \nabla f(\boldsymbol{y})\| \leq L_g \|\boldsymbol{x} - \boldsymbol{y}\|.$$

**Lemma 3.** *Suppose that Assumption 1 holds. Let $\{\boldsymbol{w}\}$ be a sequence generated by Algorithm 1. If*

$$m > 64k\vartheta/\varepsilon^4$$

*then*

$$\|\boldsymbol{p}_t - \boldsymbol{p}_t^N\| \leq \frac{1}{\lambda}(U_{Nys} + c_1\|\boldsymbol{g}_t\|^\gamma)\|\boldsymbol{p}_t\|,$$

*with probability at least $1 - \varrho$, where $U_{Nys}$ is an upper bound of $\|\boldsymbol{H}_t - \boldsymbol{N}_t\|$ given in Theorem 1. Moreover, if $\mathrm{rank}(\boldsymbol{M}) = \mathrm{rank}(\boldsymbol{H})$, then*

$$\frac{\|\boldsymbol{p}_t - \boldsymbol{p}_t^N\|}{\|\boldsymbol{p}_t\|} \leq \frac{c_1}{\lambda}\|\boldsymbol{g}_t\|^\gamma,$$

*with probability at least $1 - \varrho$ given in Theorem 1.*

*Proof.* Let direction of the Newton's method be $\boldsymbol{p}_t^N = -\nabla^2 f(\boldsymbol{w}_t)^{-1}\boldsymbol{g}_t$ and regularized Nyström direction is $\boldsymbol{p}_t = -(\boldsymbol{N}_t + \rho_t\boldsymbol{I})^{-1}\boldsymbol{g}_t$. Since $f$ is strongly convex, $\lambda_{\min}(\nabla^2 f(\boldsymbol{w})) \geq \lambda$, let $\nabla^2 f(\boldsymbol{w}) = \boldsymbol{H}$. Then we have $\|\boldsymbol{H}_t^{-1}\| \leq \frac{1}{\lambda}$ for $t > 0$. Next the distance between the directions can be given as:

$$
\begin{aligned}
\|\boldsymbol{p}_t - \boldsymbol{p}_t^N\| &= \|\boldsymbol{H}_t^{-1}(\boldsymbol{H}_t\boldsymbol{p}_t + \boldsymbol{g}_t)\| \\
&= \|\boldsymbol{H}_t^{-1}(\boldsymbol{H}_t - (\boldsymbol{N}_t + \rho_t\boldsymbol{I}))\boldsymbol{p}_t\| \\
&\leq \|\boldsymbol{H}_t^{-1}\|\|(\boldsymbol{H}_t - (\boldsymbol{N}_t + \rho_t\boldsymbol{I}))\boldsymbol{p}_t\| \\
&= \|\boldsymbol{H}_t^{-1}\|\|(\boldsymbol{H}_t - \boldsymbol{N}_t)\boldsymbol{p}_t - (\rho_t\boldsymbol{I})\boldsymbol{p}_t\| \\
&\leq \|\boldsymbol{H}_t^{-1}\|\|(\boldsymbol{H}_t - \boldsymbol{N}_t)\boldsymbol{p}_t\| + \|\boldsymbol{H}_t^{-1}\|\|\rho_t\boldsymbol{p}_t\| \\
&\leq \|\boldsymbol{H}_t^{-1}\|\|\boldsymbol{H}_t - \boldsymbol{N}_t\|\|\boldsymbol{p}_t\| + c_1\|\boldsymbol{H}_t^{-1}\|\|\boldsymbol{g}_t\|^\gamma\|\boldsymbol{p}_t\|
\end{aligned}
\tag{18}
$$

- **case a)** In this case, we discuss the distance $\|\boldsymbol{p}_t - \boldsymbol{p}_t^N\|$, when $m > 64k\vartheta/\varepsilon^4$ (Theorem 1) or $\mathrm{rank}(\boldsymbol{M}_t) < \mathrm{rank}(\boldsymbol{H}_t)$.

  Using Theorem 1 in the (18), we get

$$
\begin{aligned}
\|\boldsymbol{p}_t - \boldsymbol{p}_t^N\| &\leq \|\boldsymbol{H}^{-1}\|\|\boldsymbol{H}_t - \boldsymbol{N}_t\|\|\boldsymbol{p}_t\| + c_1\|\boldsymbol{H}_t^{-1}\|\|\boldsymbol{g}_t\|^\gamma\|\boldsymbol{p}_t\| \\
&\leq \frac{1}{\lambda}(U_{Nys} + c_1\|\boldsymbol{g}_t\|^\gamma)\|\boldsymbol{p}_t\|
\end{aligned}
$$

  where $\|\boldsymbol{H}_t^{-1}\| \leq \frac{1}{\lambda}$, and $\|\boldsymbol{H}_t - \boldsymbol{N}_t\| \leq U_{Nys}$.

- **case b)** For this case, we obtain a result when $\mathrm{rank}(\boldsymbol{M}) = \mathrm{rank}(\boldsymbol{H})$.

  Using the Theorem 2 in (18), we get

$$
\begin{aligned}
\|\boldsymbol{p}_t - \boldsymbol{p}_t^N\| &\leq \|\boldsymbol{H}^{-1}\|\|\boldsymbol{H}_t - \boldsymbol{N}_t\|\|\boldsymbol{p}_t\| + c_1\|\boldsymbol{H}_t^{-1}\|\|\boldsymbol{g}_t\|^\gamma\|\boldsymbol{p}_t\| \\
&= c_1\|\boldsymbol{H}_t^{-1}\|\|\boldsymbol{g}_t\|^\gamma\|\boldsymbol{p}_t\|
\end{aligned}
$$

  Hence, we get

$$\frac{\|\boldsymbol{p}_t - \boldsymbol{p}_t^N\|}{\|\boldsymbol{p}_t\|} \leq c_1\|\boldsymbol{H}^{-1}\|\|\boldsymbol{g}_t\|^\gamma \leq \frac{c_1}{\lambda}\|\boldsymbol{g}_t\|^\gamma$$

  where $\|\boldsymbol{H}_t^{-1}\| \leq \frac{1}{\lambda}$.

This completes the proof. □

*Remark* 3. $\boldsymbol{H}$ may not be the full rank matrix if $f$ is not strongly convex function. Then disregarding Assumption 1 for case (b) in above lemma holds for $d = m$ if $f$ strongly convex function and may be $m < d$ if $f$ not strongly convex function.

### 5.3 Linear convergence

Next, we discuss a lemma related to search direction to obtain the linear convergence.

**Lemma 4.** *Let $\boldsymbol{p}_t$ be a descent direction of Algorithm 1 at iteration $t$, then*

$$\boldsymbol{g}_t^\top \boldsymbol{p}_t \leq -\rho_t \|\boldsymbol{p}_t\|^2.$$

*Proof.* Let $\boldsymbol{p}_t = -(\boldsymbol{N}_t + \rho_t \boldsymbol{I})^{-1}\boldsymbol{g}_t$ be a search direction. Next, consider

$$\begin{aligned}
-\boldsymbol{g}_t^\top \boldsymbol{p}_t &= \boldsymbol{g}_t^\top (\boldsymbol{N}_t + \rho_t \boldsymbol{I})^{-1}\boldsymbol{g}_t \\
&= ((\boldsymbol{N}_t + \rho_t \boldsymbol{I})^{-1}\boldsymbol{g}_t)^\top (\boldsymbol{N}_t + \rho_t \boldsymbol{I})(\boldsymbol{N}_t + \rho_t \boldsymbol{I})^{-1}\boldsymbol{g}_t \\
&= \boldsymbol{p}_t (\boldsymbol{N}_t + \rho_t \boldsymbol{I})\boldsymbol{p}_t \\
&\geq \rho_t \|\boldsymbol{p}_t\|^2,
\end{aligned}$$

where the last inequality comes from the fact that $\boldsymbol{N}_t$ is positive semidefinite.
This completes the proof. □

Finally, in the next theorem, we prove the linear convergence.

**Theorem 3.** *Suppose that Assumption 1 - 4 hold. Let $\{\boldsymbol{w}\}$ be a sequence generated by Algorithm 1 and $\boldsymbol{w}_*$ be the optimal point. Then there exists $0 < \xi < 1$, with probability at least $1 - 2\exp(-m)$, we have*

$$f(\boldsymbol{w}_{t+1}) - f(\boldsymbol{w}_*) \leq \xi \left(f(\boldsymbol{w}_t) - f(\boldsymbol{w}_*)\right).$$

*where*

$$\xi = \left(1 - 4\beta(1-\beta)\frac{m\lambda^2\rho_t}{L_g(\mathcal{C}dL_g^2 + m\lambda\rho_t)}\right).$$

*Proof.* Since $\nabla f$ is Lipschitz continuous, we have

$$\begin{aligned}
f(\boldsymbol{w}_{t+1}) &\leq f(\boldsymbol{w}_t) + \boldsymbol{g}_t^\top(\boldsymbol{w}_{t+1} - \boldsymbol{w}_t) + \frac{L_g}{2}\|\boldsymbol{w}_{t+1} - \boldsymbol{w}_t\|^2 \\
&= f(\boldsymbol{w}_t) + \eta_t \boldsymbol{g}_t^\top \boldsymbol{p}_t + \frac{\eta_t^2 L_g}{2}\|\boldsymbol{p}_t\|^2
\end{aligned}$$

Let $u^2 = -\boldsymbol{g}_t^\top \boldsymbol{p}_t$, then using Lemma 4, we have $\|\boldsymbol{p}_t\|^2 \leq -\frac{\boldsymbol{g}_t^\top \boldsymbol{p}_t}{\rho_t} = \frac{u^2}{\rho_t}$, and we get

$$\begin{aligned}
f(\boldsymbol{w}_{t+1}) &\leq f(\boldsymbol{w}_t) + \eta_t \boldsymbol{g}_t^\top \boldsymbol{p}_t + \frac{\eta_t^2 L_g}{2}\|\boldsymbol{p}_t\|^2 \\
&\leq f(\boldsymbol{w}_t) + \eta_t(-u^2) + \frac{\eta_t^2 L_g}{2\rho_t}u^2 \\
&= f(\boldsymbol{w}_t) - \eta_t \left(1 - \frac{\eta_t L_g}{2\rho_t}\right)u^2
\end{aligned}$$

Hence the exit condition of backtracking line search $f(\boldsymbol{w}_t + \eta_t \boldsymbol{p}_t) \leq f(\boldsymbol{w}_t) + \beta\eta_t \boldsymbol{g}_t^\top \boldsymbol{p}_t$ satisfies if we take

$$\left(1 - \frac{\eta_t L_g}{2\rho_t}\right) = \beta,$$

and step size $\eta_t = 2(1-\beta)\rho_t/L_g$. Therefore, it stops when $\eta_t \geq 2\rho_t/L_g$ and we have

$$f(\boldsymbol{w}_{t+1}) \leq f(\boldsymbol{w}_t) - 2\beta(1-\beta)\frac{\rho_t}{L_g}u^2 \tag{19}$$

Since $u^2 = -\boldsymbol{g}_t^\top \boldsymbol{p}_t$, and from Lemma 2,

$$u^2 = -\boldsymbol{g}_t^\top \boldsymbol{p}_t = \boldsymbol{g}_t^\top (\boldsymbol{N}_t + \rho_t \boldsymbol{I})^{-1} \boldsymbol{g}_t \geq \frac{m\lambda}{\mathcal{C}dL_g^2 + m\lambda\rho_t}\|\boldsymbol{g}_t\|^2.$$

Hence, by (19) we get

$$f(\boldsymbol{w}_{t+1}) \leq f(\boldsymbol{w}_t) - 2\beta(1-\beta)\frac{m\lambda\rho_t}{L_g(\mathcal{C}dL_g^2 + m\lambda\rho_t)}\|\boldsymbol{g}_t\|^2. \tag{20}$$

Subtracting $f(\boldsymbol{w}_*)$ from both sides of (20), and from strong convexity of $f$, we have $\|\boldsymbol{g}_t\|^2 \geq 2\lambda(f(\boldsymbol{w}_t) - f(\boldsymbol{w}_*))$, which implies

$$f(\boldsymbol{w}_{t+1}) - f(\boldsymbol{w}_*) \leq f(\boldsymbol{w}_t) - f(\boldsymbol{w}_*) - 4\beta(1-\beta)\frac{m\lambda^2\rho_t}{L_g(\mathcal{C}dL_g^2 + m\lambda\rho_t)}(f(\boldsymbol{w}_t) - f(\boldsymbol{w}_*))$$

$$= \left(1 - 4\beta(1-\beta)\frac{m\lambda^2\rho_t}{L_g(\mathcal{C}dL_g^2 + m\lambda\rho_t)}\right)(f(\boldsymbol{w}_t) - f(\boldsymbol{w}_*)).$$

This completes the proof. □

### 5.4 Closeness to the Hessian inverse

In this subsection, we discuss the closeness of the inverse of regularized Nyström approximation with the Hessian inverse. Let $\boldsymbol{H}$ be the Hessian of the objective function (1) and we consider the regularized Newton's method regularized by any $\rho > 0$. Then, the inverse of Hessian of is given by $(\boldsymbol{H} + \rho\boldsymbol{I})^{-1}_{\boldsymbol{w}} = (\nabla^2 f(\boldsymbol{w}) + \rho\boldsymbol{I})^{-1}$ at $\boldsymbol{w}$. Let the regularized Nyström at $\boldsymbol{w}$ be given by $(\boldsymbol{Z_w}\boldsymbol{Z_w}^\top + \rho\boldsymbol{I})^{-1}$. The distance of the regularized inverse matrix is then given as

$$\|(\boldsymbol{Z_w}\boldsymbol{Z_w}^\top + \rho\boldsymbol{I})^{-1} - (\boldsymbol{H} + \rho\boldsymbol{I})^{-1}_{\boldsymbol{w}}\| \leq \frac{\|\boldsymbol{J_w}\|}{\rho(\|\boldsymbol{J_w}\| + \rho)}, \tag{21}$$

where $0 < \|\boldsymbol{J_w}\| = \|\boldsymbol{H} - \boldsymbol{Z_w}\boldsymbol{Z_w}^\top\| \leq \|\boldsymbol{H} - \boldsymbol{H}_k\| + \varepsilon\sum_{i=1}^d(\boldsymbol{H}_{ii})^2$; which follows from (4), whereas (21) follows from (Frangella et al., 2021, Proposition 3.1).

Note that the rank of Hessian can be possibly $r$ when the objective function is not $\ell_2$ regularized.

## 6 Stochastic variant of the regularized Nyström gradient method

In this section, we discuss the stochastic variant of the Nyström gradient. In the context of machine learning, it is usual to work with a large number of samples, making it computationally challenging to compute the full gradient at every iteration. To address this challenge, we employ the stochastic gradient with the Nyström approximation. In this stochastic variant of the NGD, we compute the mini-batch stochastic gradient at every iteration and compute the regularized Nyström $\boldsymbol{N}_\tau + \rho_\tau \boldsymbol{I}$, once per epoch with $\rho_\tau = c_1\|\boldsymbol{g}_{t-1,\tau}\|^\gamma$ [9]. We call this variant NSGD. Furthermore, we use diminishing step size $\eta_t$ for the stochastic variant NSGD.

---

[9]that the regularizer $\rho_\tau$ is stochastic gradient and not full gradient. We update the $\rho_\tau$ in the beginning of the epoch $\tau$, with $\nabla f_{\mathcal{B}}(\boldsymbol{w}_t)$ of $(\tau - 1)$th epoch.

Table 2: Search direction and $\gamma$ in for NSGD

| Proposed methods | Regularizer $\rho$ (Value of $\gamma$) | Search direction |
|---|---|---|
| NSGD | $\rho_\tau = c_1\|\boldsymbol{g}_{t-1,\tau}\|^\gamma$ ($\gamma = 1/2$) | $\boldsymbol{p}_{t-1} = (\boldsymbol{N}_\tau + \rho_\tau)^{-1}\boldsymbol{g}_{t-1,\tau}$ |

---

**Algorithm 2** NysReg-Stochastic gradient: NSGD Algorithm

**Parameters:** Update frequency $\ell$ and initial step size $\eta_0$

1: **Initialize** $\boldsymbol{w}_0, \tau = 1$
2: **for** $t = 1, 2, \ldots$ **do**
3:     randomly pick batch $\mathcal{B} \sim \{1, \ldots, n\}$
4:     $\boldsymbol{g}_{t-1,\tau} = \nabla f_{\mathcal{B}}(\boldsymbol{w}_{t-1})$
5:     **if** $(t-1) \bmod \ell = 0$ **then**
6:         randomly pick indices set $\boldsymbol{\Omega} \subseteq \{1, 2, \ldots, d\}$ such that $m = |\boldsymbol{\Omega}|$
7:         compute $\boldsymbol{C}_\tau$ ($\boldsymbol{\Omega}$ columns of the Hessian) at $\boldsymbol{w}_{t-1}$
8:         compute $\boldsymbol{Z}_\tau$ using (7) and compute $\rho_\tau$
9:         $\boldsymbol{Q}_\tau = \frac{1}{\rho_\tau^2}\boldsymbol{Z}_\tau(\boldsymbol{I} + \frac{1}{\rho_\tau}\boldsymbol{Z}_\tau^T\boldsymbol{Z}_\tau)^{-1}$
10:      $\tau = \tau + 1$
11:     **end if**
12:     Compute $\boldsymbol{p}_{t-1}$ using (11)
13:     $\boldsymbol{w}_t = \boldsymbol{w}_{t-1} - \eta_t\boldsymbol{p}_{t-1}$
14: **end for**

---

# 7 Numerical experiments

In this section, we demonstrate the numerical results for the proposed algorithms explained in the previous sections.

First, we discuss the experiment setup for the numerical experiments. We performed Figure experiments on MATLAB R2018a on Intel(R) Xeon(R) CPU E7-8890 v4 @ 2.20GHz with 96 cores and Figure on MATLAB R2019a on Intel(R) Xeon(R) CPU E5-2620 v4 @ 2.10GHz with 32 cores. We implemented the existing and proposed methods in MATLAB using the SGDLibrary (Kasai, 2017). We solve on standard learning problems, that is, the $\ell_2$-logistic regression:

$$\min_w F(w) = \frac{1}{n}\sum_{i=1}^{n}\log\left[1 + \exp(-b_i a_i^T w)\right] + \frac{\lambda}{2}\|w\|^2,$$

where $a_i \in \mathbb{R}^d$ is feature vector and $b_i \in \{\pm 1\}$ is target label of the $i$-th sample, and $\lambda$ is a $\ell_2$ regularizer. We evaluated the numerical experiments on benchmark datasets given in Table 3. The datasets are binary classification problems and all datasets are available on LIBSVM Chang & Lin (2011). We demonstrate the performance of the proposed and existing methods on the $\ell_2$-regularized logistic regression problem. We optimize the constant $c_1$ in regularizer $\rho_t = c_1\|\boldsymbol{g}_t\|^\gamma$ using a grid search $c_1 \in \{10^0, 10^{-1}, 10^{-2}, 10^{-3}\}$. For each method, the best-performing model was selected based on the minimum cost error on the training. For the numerical experiments conducted on $\ell_2$-regularized squared SVM, see Appendix section B.

Table 3: Details of the datasets used in the experiments

| Dataset | Dim | Train | Test | Density |
|---|---|---|---|---|
| $adult^1$ | $123 + 1$ | 32,561 | 16,281 | 0.1128 |
| $gisette^1$ | $5,000 + 1$ | 6,000 | 1,000 | 0.9910 |
| $epsilon^1$ | $2,000 + 1$ | 50,000 | 50,000 | 1 |
| $real\text{-}sim^1$ | $20,958 + 1$ | 57,909 | 14,400 | 0.0024 |
| $w8a^1$ | $300 + 1$ | 49,749 | 14,951 | 0.0388 |

---

[1]Available at LIBSVM (Chang & Lin, 2011) `https://www.csie.ntu.edu.tw/cjlin/libsvm/`

First we study the performance of NGD, NGD1 and NGD2 to see the behaviour of different $\rho = c_1\|\boldsymbol{g}_t\|^\gamma$, where $\gamma = 1/2$ for NGD, $\gamma = 1$ for NGD1, and $\gamma = 2$ for NGD2. We computed the $\ell_2$-regularized *logistic regression* with $\lambda = 10^{-5}$ on the *ijcnn1* and *adult* datasets.

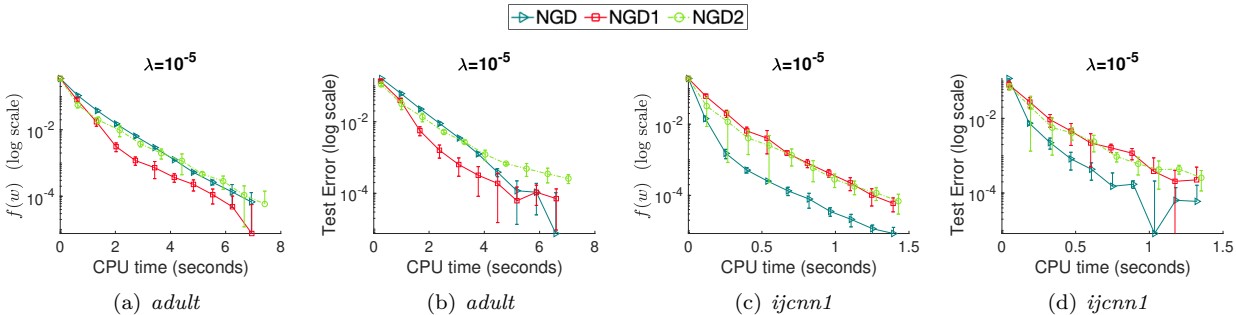

Figure 1: First two figures(from left) shows the experiments on *adult* for $m = 25$ and last two figures shows the experiments on *ijcnn1* for $m = 5$. (a) and (c) shows the cost with respect to CPU time. (b) and (d) shows the test error with respect to CPU time.

Figure 1 shows the training cost and test error with CPU time for *adult* and *ijcnn1*. Moreover, it shows that NGD1 outperforms NGD and NGD2 for *adult* dataset and NGD outperforms NGD1 and NGD2 for *ijcnn1* dataset. Therefore, in the next subsection, we consider NGD and NGD1 to compare the behavior with varying numbers of selected columns.

## 7.1 Comparison of strength for varying numbers of selected columns

In this subsection, we demonstrate the comparison of various sketch sizes (no. of selected columns) for high-density data *gisette* and sparse data *w8a* on *logistic regression* with $\lambda = 10^{-5}$ to observe the robustness of the proposed methods. We keep the same $c_1$ in $\rho_t$ for each dataset to compare the different numbers of selected columns. Figure 2 shows the numerical performance for the *gisette* dataset and computed the NGD1 for $m = 50(1\%), 250(5\%), 500(10\%)$ and $m = 1000(20\%)$. As shown in Figure 2, due to high density only $m = 250(5\%)$ of columns are sufficient to get the minimum value of the objective function within the comparative CPU time. Also, similar behavior can be observed in the test error as well. When $m = 1000$, the decrement in the value of the gradient norm surpasses all cases of $m < 1000$. Additionally, $m = 250$ and $m = 500$ perform a similar reduction in the value of the norm of the gradient.

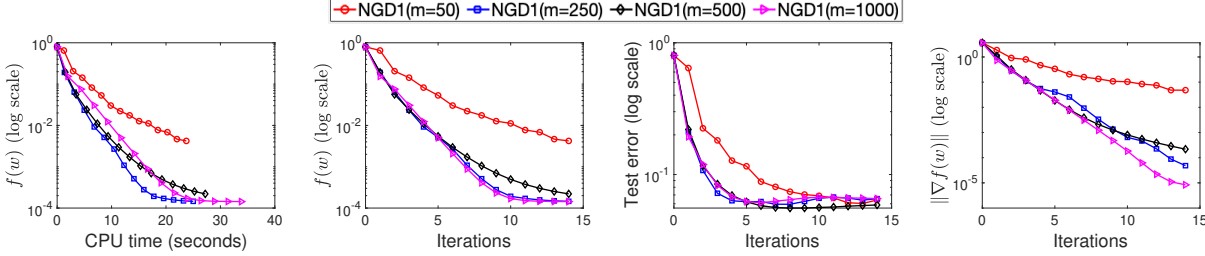

Figure 2: Column comparison on *gisette* dataset

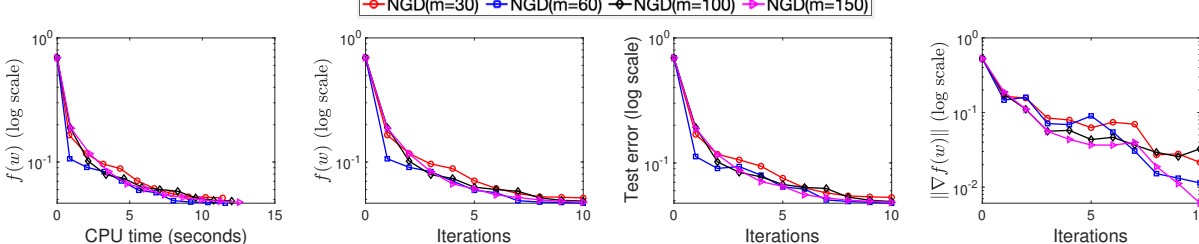

Figure 3: Column comparison on *w8a* dataset

Figure 3 shows the numerical performance on the *w8a* dataset and computed NGD for $m = 30(10\%), 60(20\%), 100(33\%)$ and $m = 150(50\%)$. Figure 3 shows that due to sparse data, it requires picking more numbers columns to obtain the minimum value of the objective function in the comparative CPU time. All cases of $m$ exhibit almost similar test error. When $m = 150$ and $m = 100$, the decrement in the value of the norm of gradient is comparable, whereas for the cases $m = 30$ and $m = 60$, it does not decrease significantly.

## 7.2 Comparison with randomized subspace Newton

In this subsection, we compare the NGDs with the randomized subspace Newton (RSN) (Gower et al., 2019). RSN computes the iterate with $\boldsymbol{w}_{t+1} = \boldsymbol{w}_t - (1/L)\boldsymbol{S}_t(\boldsymbol{S}_t^\top \boldsymbol{H}_t \boldsymbol{S}_t)^\dagger \boldsymbol{S}_t^\top \boldsymbol{g}_t$ with a sketch matrix $\boldsymbol{S}_t \in \mathbb{R}^{d \times m}$. To have a fair comparison of the subspace Newton, we compute the RSN with the Armijo's rule with backtracking line search (instead of $1/L$) and compute RSN exactly as given in (Gower et al., 2019, definition 4) for generalized linear models. Also, we keep the same value of $m$ for both NGDs and RSN. We compute the *logistic regression* with $\lambda = 10^{-5}$. We compare NGDs and RSN in Figure 4 for *realsim* data with $m = 2000$, Figure 5 for *gisette* data $m = 250$, and Figure 6 for *w8a* data with $m = 30$. As shown in Figure 4 to 6, RSN is unable to outperform the proposed methods. For the *realsim* data, as shown in the Figure 4, NGD1 outperforms all methods in terms of achieving the minimum cost and NGD2 outperforms all methods in terms of test error. For the *gisette* data, In Figure 5, NGD1 outperforms all methods in terms of achieving minimum cost and test error. Finally, for *w8a* dataset, Figure 6, NGD outperforms all methods in terms of achieving minimum cost and NGD1 outperforms at the later stage in terms of test error. In conclusion, it is observable that the Nyström approximation is better than the approximation of RSN because RSN only captures a limited set of $m^2$ elements from the Hessian, whereas Nyström captures a substantially larger set of $dm$ elements of the Hessian. This makes Nyström approximation more comprehensive and accurate of the Hessian matrix.

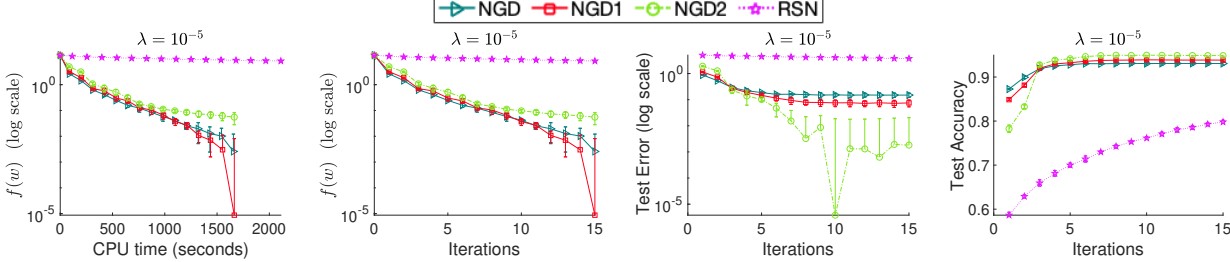

Figure 4: Comparison with RSN for *realsim* dataset with $m = 2000$

## 7.3 Comparison of Newton Sketch and Nyström approximation

In this subsection, we compare the NGDs with the Newton sketch(NS) (Pilanci & Wainwright, 2017). As explained in Section 4.2, the proposed method can be represented as the NS method with certain structure modifications. Hence, we compare the raw Nyström with the Hessian approximation of NS in terms of

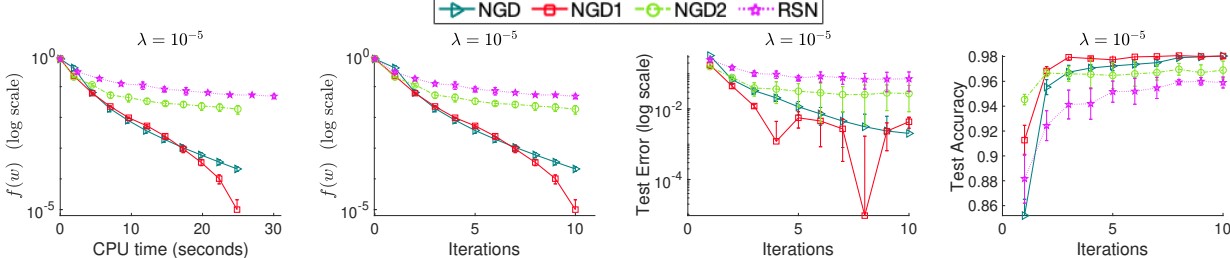

Figure 5: Comparison with RSN for *gisette* dataset with $m = 250$

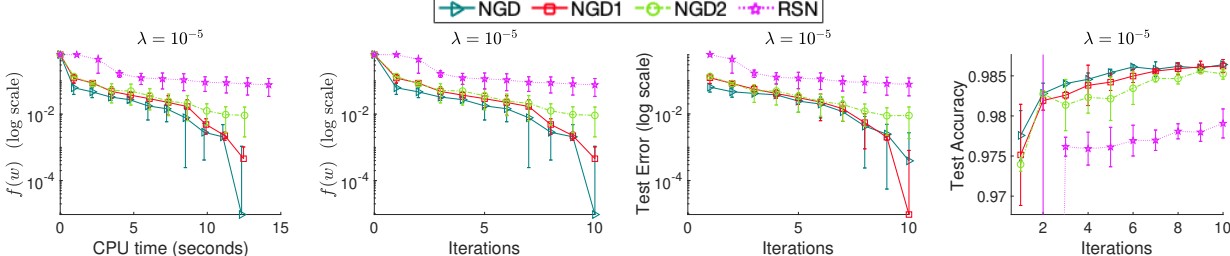

Figure 6: Comparison with RSN for *w8a* dataset with $m = 30$

closeness with the Hessian. NS computes the Hessian approximation as $(\nabla^2 f(\boldsymbol{w})^{1/2})^\top \boldsymbol{P}^\top \boldsymbol{P}(\nabla^2 f(\boldsymbol{w})^{1/2})$, and it is important to note that the $\boldsymbol{P} \in \mathbb{R}^{m \times n}$, where $n$ is the number of samples and $m$ is the sketch size. In this comparison, we keep the same value of $m$ for both Nyström and NS. In Figure 7 we conduct numerical experiments on *w8a, realsim* and *gisette* datasets. We provide the comparison of norm difference with Hessian and its CPU time as $m$ increases. We conduct these numerical experiments on the *logistic regression*. It is important to note that we use the *logistic regression* for the *realsim,* and *gisette* without $\ell_2$ regularization. For the *w8a* dataset, we keep the $\ell_2$-regularized *logistic regression* with $\lambda = 10^{-5}$. Hence the rank of $\boldsymbol{H}$ for *w8a* data is full. In Figure 7, (a) and (d) show the performance on *w8a* dataset, (b) and (e) show the performance on the *realsim* dataset, and (c) and (f) shows the performance on the *gisette* dataset. The top row shows the CPU time of computing the Nyström approximation and Newton sketch and the bottom row shows the distance with Hessian as $m$ increases, where $\boldsymbol{H}$ is the Hessian.

As shown in Figure 7 (a) and (d), Nyström approximation outperforms the Newton sketch as $m$ increases with the less CPU time for *w8a* in lesser CPU time compared to NS. Similarly, in the Figure 7(b) and (e), the Nyström approximation can approach the Hessian as $m$ increases, specifically, after $m = 8000$. Since, Nyström involves the inverse of $m \times m$ matrix, it takes more CPU time after $m = 5000$. Whereas in Figure 7(c) and (f), the norm of distance between Hessian and Nyström decreases significantly when $m \approx 1000$ and takes more CPU time after $m \approx 1000$ compared to NS. However, we do not need to compute Nyström for large number of $m$, as we have seen in Figure 2 and 3 that about 5% to 15% of $d$ can give sufficient decrease in the objective function.

From performance illustrated in Figure 7, two significant observations can be made. Firstly, one can observe in Figure 7(d) that the Theorem 2 pertaining Nyström bound of exactness holds true practically and becomes almost zero as the number of columns $m$ covers the all of the columns (*i.e.,* rank of $\boldsymbol{H}$). Secondly, it is worth noting that the random matrix in the Newton sketch $\boldsymbol{P} \in \mathbb{R}^{m \times n}$ depends on the $n$ which is usually larger than dimension $d$, whereas Newton sketch (Pilanci & Wainwright, 2017) usually requires thick random matrix as compare to the thin random matrix $\boldsymbol{S}$ of Nyström approximation.

## 7.4 Comparison with existing deterministic methods

We compared the proposed methods NGD, NGD1, and NGD2 with the existing classical first-order gradient descent, and the *state of the art* second-order Hessian approximation method L-BFGS method Liu &

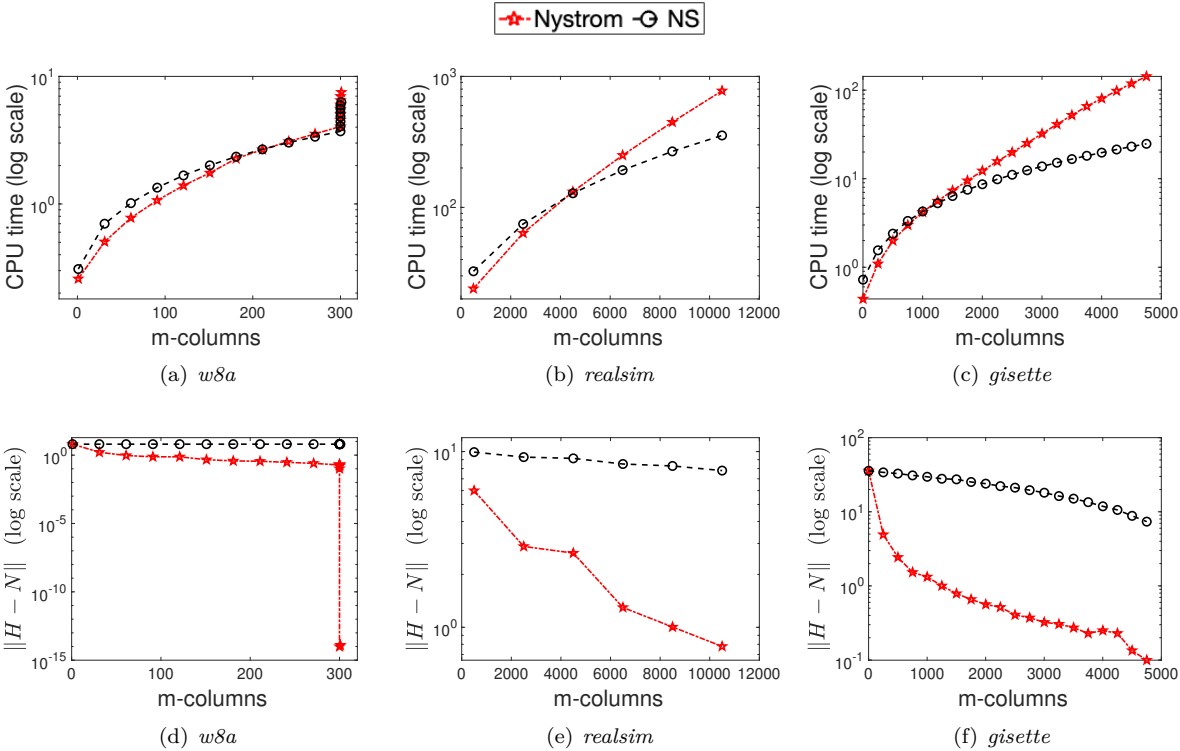

Figure 7: Comparison between Nyström and Newton sketch

Nocedal (1989). The memory used in the L-BFGS method was set to 20. We report the training cost on the training dataset and testing set (test error) for iteration and CPU time cost per iteration. Also, we show the norm of the gradient with respect to iterations. Figure 8 shows the performance of experiments on *logistic regression* with $\lambda = 10^{-5}$ on *giestte* dataset with $m = 500$. As shown in Figure 8, NGD1 outperforms all other methods in terms of both CPU time and iterations in terms of both training cost and the norm of gradient. L-BFGS takes more CPU time compared to all variants of NGDs till the cost of $10^{-5}$. Also, GD shows improvements after the 20th iteration and outperforms in terms of the test error and NGD2 shows some increment in the test accuracy after the 30th iteration. In Figure 9, we conduct the experiments on *logistic regression* with $\lambda = 10^{-5}$ on *epsilon* dataset with $m = 200$. Figure 9 shows that the NGDs are performing almost similarly and outperform the L-BFGS and GD in terms of the training cost, testing error, and test accuracy. Also, NGD and NG1 outperform all of the methods in terms of the norm of gradient.

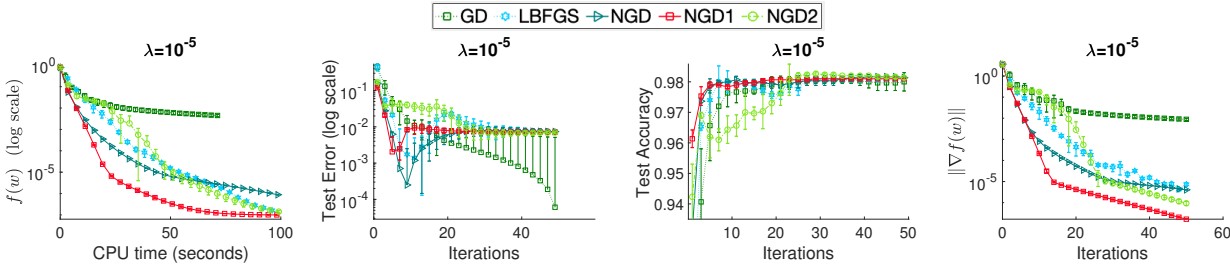

Figure 8: Experiments on the *gisette* dataset with $m = 500$.

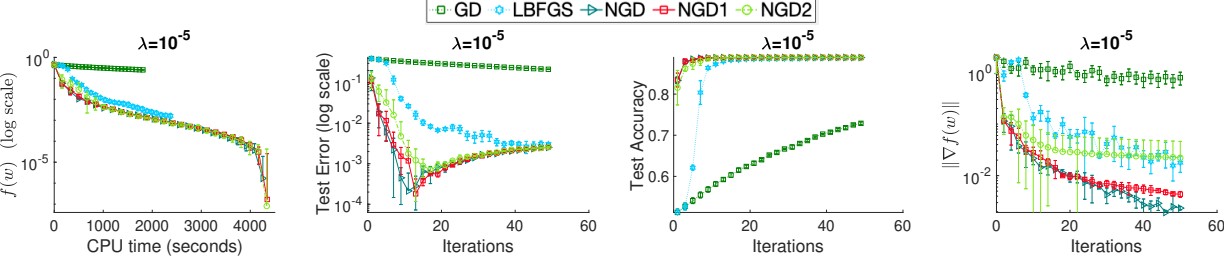

Figure 9: Comparison for *epsilon* dataset with $m = 200$

## 7.5 Numerical experiments for stochastic regularized Nyström gradient

We compare the proposed stochastic variant NSGD with stochastic gradient descent method and stochastic second order approximation optimization methods, namely, SVRG-LBFGS (Kolte et al., 2015), SVRG-SQN (Moritz et al., 2016), and SQN (Byrd et al., 2016). The memory used in the L-BFGS method was set to 20, which is a commonly used value (Kolte et al., 2015; Byrd et al., 2016). Figure 10 shows that NSGD

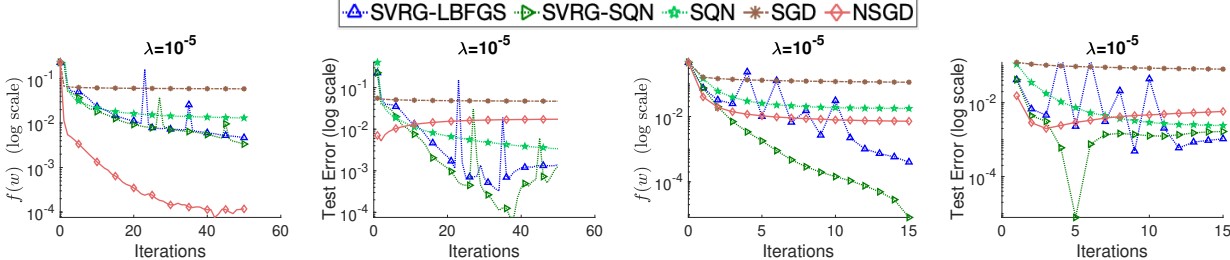

Figure 10: First two from left shows the experiments on *a8a* dataset and two from right shows the experiments on *epsilon* dataset

outperforms existing methods in terms of the training cost for *a8a* dataset. However, it could not achieve a better test error compared to SVRG-SQN and SVRG-LBFGS. Moreover, SVRG-SQN outperforms NSGD and other existing methods in terms of both training cost and test error for *epsilon* dataset.

## 7.6 Numerical experiments for deep learning

We also evaluated the performance of the Nyström SGD on the well-known deep models on the Imagenet dataset.

**Experimental Setup:** We compared our method with the first-order methods SGD and the well-known approximate second-order method KFAC Martens & Grosse (2015) on ResNet152 He et al. (2016) and EfficientNet Tan & Le (2019) models.

For Nyström SGD, we used $\rho = 0.1$ and fixed the $m = \log_2 |\boldsymbol{w}|$, where $|\boldsymbol{w}|$ is the number of parameters in the respective model. We used a batch size of 128. We used a random sample of size of $\min\{6400, n \times 0.01\}$ to compute the partial Hessian $\boldsymbol{C}$ for Nyström SGD. The update frequency used to re-estimate the preconditioner in KFAC and its variants is set to 200, as used in their experiments. The ImageNet results were computed on a Quadro RTX 8000 GPU.

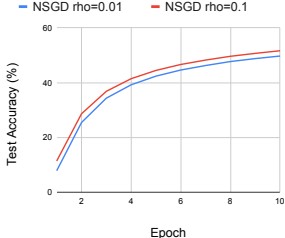

Figure 11: Effect of the $\rho$ on the test accuracy for ResNet18 on imagenet dataset.

**Results:** Figure 12 presents the results of the ResNet152 and EfficientNet on the ImageNet dataset. The proposed method outperformed both the SGD and

KFAC for both the models in terms of training loss as well as test accuracy, showing the better optimization and generalization ability of the trained models. Table 4 shows the computational time comparison of methods. The per update computational time of the Nyström SGD on ResNet152 is 1.703 seconds which is slightly slower than SGD and KFAC. To further speed up the Nyström SGD is an interesting future work. Figure 11 shows the effect of the $\rho$ parameter for ResNet18. As can be seen, the $\rho$ parameter affects the model performance. We found setting $\rho = 0.1$ performs well in practice.

Table 4: Per iteration computational time (seconds). [*For KFAC on EfficientNet with batch size 128 could not fit into memory]

| Model | Method | Batch | Update Time | Hessian |
|---|---|---|---|---|
| ResNet152 | SGD | 128 | 1.006 | - |
| | KFAC | 128 | 1.064 | - |
| | NSGD | 128 | 1.060 | 0.643 |
| EfficientNet | SGD | 128 | 0.341 | - |
| | KFAC | 64* | 1.173 | - |
| | NSGD | 128 | 0.347 | 2.620 |

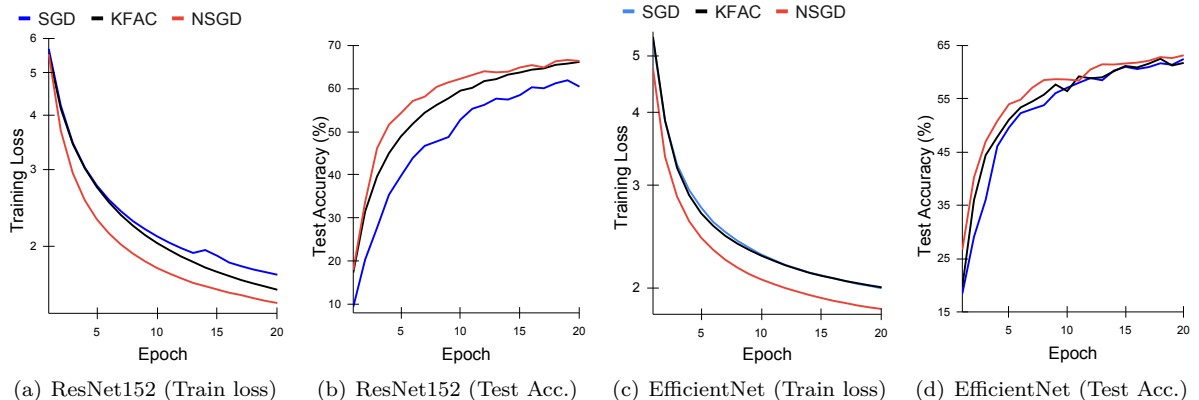

(a) ResNet152 (Train loss)  (b) ResNet152 (Test Acc.)  (c) EfficientNet (Train loss)  (d) EfficientNet (Test Acc.)

Figure 12: Results on Imagenet using ResNet152 and EfficientNet, respectively.

## 8 Application: Tumor detection

Brain MRI is the most standard test for the diagnosis of various brain diseases including tumor detection. Given the complexity of the diagnosis process, researchers are shifting towards deep neural networks. First-order optimizers are the most preferable choice in deep learning. However, with the limited sample sizes, it is difficult to train a stable and generalized model with a large number of parameters using first-order optimizers. We consider studying the *brain MRI images for brain tumor detection*. This data contains 253 MRI images where 155 cases have tumors and 98 cases are of the healthy brain. We use a transfer learning approach to detect the tumor. Transfer learning is widely used in brain MRI and biological problems where the number of samples is limited. In deep models, bottom layers perform generic tasks such as edge detection. Whereas, the top layers are task specific. Hence the common practice is to fine-tune the top layers only. Goal is to minimize the objective function

$$\min_w f(w), \quad \text{where} \quad f(w) = \sum_{i=1}^{n} f_i(w),$$

where $w \in \mathbb{R}^d$ and, $f : \mathbb{R}^d \to \mathbb{R}$, and $f_i$ is the loss function corresponding to $i^{th}$ sample is the *logistic regression* for brain tumor *classification* problem. *i.e.*, The data has $d$ dimension and $n$ samples. We propose an NGD

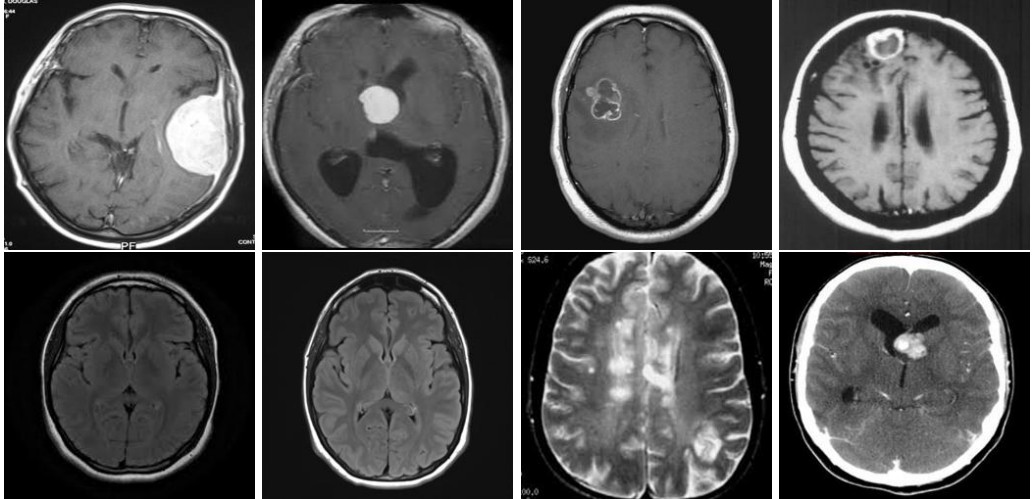

Figure 13: Sample Images from MRI dataset dat (2020), Top row: Tumor, Bottom row: Healthy

algorithm for fine-tuning the top layers of pre-trained deep networks. Specifically, we compute a partial column Hessian of size $(d \times m)$ with $m \ll d$ uniformly randomly selected variables ($d$ is the number of parameters), then use the *Nyström method* to approximate the full Hessian matrix.

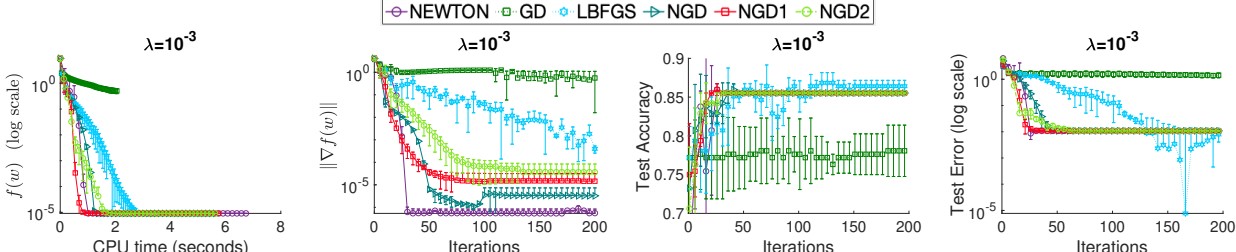

Figure 14: Comparison of NGDs with existing methods on MRI dataset

Figure 14 shows that NGD1 outperforms other methods in terms of training cost in the least CPU time. Newton's method outperforms in terms the decreasing the norm of the gradient. Additionally, all NGDs are giving competitive behavior to each other in terms of the norm of gradient. GD and L-BFGS are not able to give competitive results in terms of test accuracy and test error. Also, all NGDs and Newton's method have the upper hand in achieving better test accuracy and test error.

## 9 Summary

In this paper, we introduce the regularized Nyström method to approximate Hessian and propose both deterministic and stochastic optimization methods to solve the objective function. We present the comprehensive convergence analysis and certain results using the distance between the Hessian and Nyström approximation. Furthermore, we conducted extensive numerical experiments to evaluate the performance of the proposed methods with RSN (Gower et al., 2019), NS (Pilanci & Wainwright, 2017), and other existing first and quasi-Newton methods. From the numerical results, the proposed methods demonstrate robustness, efficiently approximating the Hessian by selecting approximately 5%(in high-density scenarios) and 15-20%(in high-sparsity scenarios) of the dimension. Moreover, we prolong the experiments to the domain of deep learning and we employ our proposed method for an application involving brain tumor detection. The results in this application highlight the promising impact of our proposed methods in real-world scenarios.

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
