# OpenReview forum: "NysReg-Gradient:~Regularized Nystr\"om-Gradient for Large-Scale Unconstrained Optimization and its Application"
_TMLR — Rejected by TMLR_

### Review · Reviewer_9hJL · 2024-03-06

**Summary Of Contributions:**

This paper studies a regularized quasi-Newton method where the Hessian is approximated by its k-rank nyström approximation. They propose to use the gradient information to get an adaptive regularization value. The authors show that with high probability, the suboptimality can decrease after one optimization step of their algorithm. I have some concerns regarding the theory (see section strenghts and weaknesses). They also try their algorithm in practice

**Audience:**

No

**Broader Impact Concerns:**

no concerns

**Claims And Evidence:**

No

**Requested Changes:**

I would like the author to:
- comment on my concerns regarding the theory (i.e. can you show a convergence rate with the relative condition number similar to Gower et al. 2019)
- Comment on my concern about Lemma 2
- Provide a comparison with the baseline in terms of convergence speed (e.g. SOTA newton-cg or variance reduction method for logistic regression used in sklearn)
- clarify the writing (See my comments above)

**Strengths And Weaknesses:**

# Strength:

The authors implemented their method in many experimental settings.

# Weakness:

## Significance of the Theory

From a high level perspective I am quite concerned about the significance of the theoretical results presented in this paper. In particular, the main result (theorem 3) do not leverage the fact that the sketching matrix S is random in order to provide an estimation of the Hessian. The proof of Theorem 3 is valid for any fixed sketching matrix as long at $\|SS^\top\| \leq Cd/m$. It lacks significance since:
- Theorem 3 is not a global convergence results
- The condition number appearing is worse than the one appearing for gradient descent. Unlike in other analysis of quasi-Newton method like Gower et al. 2019 or Karimireddy et al. 2018 that make appear the **relative** smoothness and strong convexity (relative to the metric induced by the Hessian)

### About Lemma2

I have a concern regarding the proof of Lemma 2. In the paper it is mentioned that “since f is strongly convex and $m << d$ we have $S^\top HS \succeq \lambda I_d$. I believe this statement is wrong in general (i.e. we need more assumptions on S). For instance if we were to take S = 0 it is clear that this statement is not true (any S close to 0 would also make the statement wrong)

I believe that this could be fixed as on could instead consider upper bounding the eigenvalues of $S(S^\top HS)^\dagger S$ whose eigenvalues are upper bounded by $1/\lambda$.

### About Section 6
Section 6 is quite small, the author extend their method to the stochastic case without really discussing the impact of this stochasticity on the optimization. I am actually very surprised that, in section 7.5 NSGD converges faster that SVRG since the latter has a linear convergence rate (and I do not see how the former would have a linear convergence rate without variance reduction). Moreover, the comparison is not doing using the time as a the x-axis which is not fair for the baselines as the cost per iteration is significantly cheaper.

## Significance of the experiments

Most of the experiments are performed on L2 logistic regression without comparing the method with a proper baseline (in terms of convergence speed). I would suggest the author to consider using the BenchOPT library [Moreau et al. 2022] to properly benchmark their method against the baseline for logistic regression.

## Writing:
The writing can be significantly improved. I found some sections quite confusing

### About the random sketching matrix $S$:
Nyström approximation corresponds to equation (2) or (6). A connection is made with the sketching matrix S = WD. However, this matrix S has a specific form it is the product of the zero-one matrix W defined in (5) with a diagonal matrix with positive coefficient. But then the authors consider random matrices with Gaussian entries.

It seems to me that as long as S is of rank m we have that $(HS)(S^\top HS)^\dagger_k(HS)^\top = CM_k^\dagger C^\top$ so this would be the formulation to consider.

### Section 4

I find Algorithm 1 not very clear.

 - In order to compute C_t you need to sample S. do you sample S with Gaussian entries, or do you consider the matrix W defined in (5). If you do consider the matrix W defined in (5) what do you mean in the statement P4 “for the rest of theoretical analysis, we consider the matrix S to a a generalized random matrix given in Lemma 1”? moreover in assumption 2 you consider S_t to have i.i.d. gaussian entries.
 - p_t is not defined in algorithm !
 - line 7 there is an extra space before (7)

I find some statements in section 4.1 somewhat confusing. It is claimed that “the overall …complexity are O(dm)” but the SVD necessary to compute Z_t has a complexity O(dmk).

### Typos and inaccurate notations:
There are numerous typos and inaccurate notations. Here is a non exhaustive list
  - P3: “the following theorem shows”
  - P3: “let […] be a k-rank ($k \leq m$) is”
  - P3: let $k = rank(M_k)$ (k is not the rank of M)
  - P3: “If is always possible”
  - P4: “random trail” and “independent trail” are not defined. Did you mean random trial?
  - P4: “exactly same as the…”
  - P6-7: some matrices are not bolded (S, X in (14)) or are missing a subscript t (C in 4.1, S in Assumption 2),
  - P8 “the the”

---

### Review · Reviewer_NQwL · 2024-03-07

**Summary Of Contributions:**

This paper proposes a new quasi-Newton step algorithm for minimizing a strongly convex and smooth function. The main idea is to conduct Newton step but use Nystrom method to approximate the Hessian. The authors show that the proposed algorithm enjoys linear convergence, and the experiments show the effectiveness of the proposed algorithms.

**Audience:**

Yes

**Claims And Evidence:**

Yes

**Requested Changes:**

The are listed in the section above.

**Strengths And Weaknesses:**

Strengths:
1. The authors give a rigorous proof and show that their proposed method enjoys a linear convergence rate. Although GD can achieve the same rates in this case (strongly convex and smooth).
2. The authors conduct extensive experiments, and the results shows the effectiveness of the proposed algorithms.



Weakness:


1. I have some questions on the proof. Specifically, I am not sure if the convergence analysis (Theorem 3, which is the main theorem) is meaningful enough. After a close look, I find that the only Nystrom-approximation  property that is used in the proof is Lemma 2, that is, the Regularized matrix has bounded max and min eigenvalue. But if one uses c*I_d to replace N_t (that is, use GD, not Nys-approximation), it seems that the analysis remains that same. That is to say, 1) the authors gave several bounds about the gap between Nystrom-approximation and real Hessian, but they are not helpful at all in the convergence analysis; 2) The proof is essentially the proof of GD for strongly convex and smooth functions (which indeed get linear convergence), and is not very related to Newton (since N_t+I has both upper and lower bound, the proof reduces to the proof of GD). How Nys-GD is better than GD is not clear (in terms of convergence rate). On the other hand, GD is more efficient.


2. In terms of computational complexity (Page 6), I am not sure if the claim is true. Clearly, dkm>dk, so the computational complexity should at least be dkm. Moreover, I am not sure how good or bad it is compared to other approximation methods. It would be great if the authors can add more discussion.

3. There is no theoretical guarantees for the stochastic version of the algorithm.

4. The writing of the paper can be improved. There are many typos and grammar errors, which make the paper difficult to read. For example:
Definition 1: "Let m*m be a matrix M";
Theorem 1: "Let N_k be a k-rank is a Nystrom approximation"
Remark 3: "Then disregarding Assumption 1 for case (b) in above lemma holds for d = m if f strongly convex function and may be m < d
if f not strongly convex function." I am not sure about the meaning of the sentence.

---

### Review · Reviewer_wrh8 · 2024-03-26

**Summary Of Contributions:**

The authors propose a new Quasi-Newton type optimization method with Nystrom approximation of the Hessian and regularised by the gradient norm. The convergence proofs for strongly convex and smooth optimization are presented. Also, the closeness/inexactness of the Nystrom approximation is studied. Experimental results are presented for logistic regression with real-life datasets of reasonable size. Non-convex experiments for deep learning setup are also presented.

**Audience:**

Yes

**Claims And Evidence:**

Yes

**Requested Changes:**

I would also recommend to improve the Related work section: The recommended citations on Quasi-Newton methods:

BFGS

 1. Charles G Broyden. Quasi-Newton methods and their application to function minimisation. Mathematics of Computation, 21:368–381, 1967

2. Roger Fletcher. A new approach to variable metric algorithms. The Computer Journal, 13:317–322, 1 1970

3. Donald Goldfarb. A family of variable-metric methods derived by variational means. Mathematics of Computation, 24:23–26, 1970

4. David F Shanno. Conditioning of Quasi-Newton methods for function minimization. Mathematics of Computation, 24:647–656, 1970.

SR-1/L-SR-1:

5. Andrew R Conn, Nicholas IM Gould, and Philippe L Toint. Convergence of quasi-newton matrices generated by the symmetric rank one update. Mathematical Programming, 50:177–195, 1991.

6. H Fayez Khalfan, Richard H Byrd, and Robert B Schnabel. A theoretical and experimental study of the symmetric rank-one update. SIAM Journal on Optimization, 3:1–24, 1993.

7. Albert S Berahas, Majid Jahani, Peter Richt ́arik, and Martin Tak ́aˇc. Quasi-newton methods for machine learning: forget the past, just sample. Optimization Methods and Software, pages 1–37, 2021.

Second-order and regularised Newton methods:

8. Yurii Nesterov and Boris T Polyak. Cubic regularization of Newton method and its global performance. Mathematical Programming, 108:177–205, 2006.

9. Nikita Doikov and Yurii Nesterov. Gradient regularization of Newton method with Bregman distances. Mathematical Programming, 2023.

10. Konstantin Mishchenko. Regularized Newton method with global O(1/k^2) convergence. SIAM Journal on Optimization, 33(3):1440–1462, 2023

11. Boris Teodorovich Polyak. Newton’s method and its use in optimization. European Journal of Operational Research, 181:1086–1096, 2007.

11. Roman A Polyak. Regularized Newton method for unconstrained convex optimization. Mathematical Programming, 120:125–145, 2009.

**Strengths And Weaknesses:**

The paper has an extensive and high-level study on Nystrom approximation of the Hessian. This topic is novel and highly perspective. It allows to reduce the computational costs of the second-order methods while accelerating the convergence of first-order methods with additional second-order information. The paper is well-written and the proofs seem to be correct. The experiments are also performed for various problems including non-convex Resnet152 and EfficientNet.

 For the weaknesses, the theoretical convergence rates are not as fast as they could be but I think this is a byproduct of the method’s choice. So, it is not a problem of Nystrom approximation.

Does the proposed method have a connection with the SR-1 method as its sampled version?

---

### Decision · Action_Editor_Sevy · 2024-05-14

**Recommendation:** Reject

**Comment:**

In summary, while the paper offers some relevant contributions, they are insufficient and lack the necessary evidence to support the claims made. As a result, it does not fulfill the TMLR submission criteria, leading to the decision to reject the paper.

**Audience:**

Yes, appropriate for the audience.

**Claims And Evidence:**

The paper introduces a Nyström-Gradient method, which utilizes the Nyström approximation of the Hessian matrix for unconstrained problems, demonstrating good performance in applications.

The paper asserts the novelty of this method and its performance benchmarks. While all reviewers find the Nyström approximation of the Hessian matrix to be of interest, they deem it quite limited. The reviewers' main concern is the lack of comprehensive exploitation in terms of analysis within the paper. For instance, if the method is presented as a quasi-Newton method, the paper should theoretically demonstrate its superlinear convergence. This was a shared concern among two reviewers, which was not adequately addressed in the response and revision. Regarding the experiments, the computational advantages over methods like the conjugate gradient method are unclear, as the experiments lack thoroughness and only present a straightforward application of the proposed method.

**Resubmission Of Major Revision:**

The authors may consider submitting a major revision at a later time.